*Report*

# Bacterial lysis, autophagy and innate immune responses during adjunctive phage therapy in a child

Ameneh Khatami[1,2,*] ![iD], Ruby C Y Lin[2,3,4] ![iD], Aleksandra Petrovic-Fabijan[3] ![iD], Sivan Alkalay-Oren[5,6], Sulaiman Almuzam[1], Philip N Britton[1,2], Michael J Brownstein[7], Quang Dao[8], Joe Fackler[7], Ronen Hazan[5], Bri'Anna Horne[7], Ran Nir-Paz[6] & Jonathan R Iredell[2,3,9,**] ![iD]

## Abstract

Adjunctive phage therapy was used in an attempt to avoid catastrophic outcomes from extensive chronic *Pseudomonas aeruginosa* osteoarticular infection in a 7-year-old child. Monitoring of phage and bacterial kinetics allowed real-time phage dose adjustment, and along with markers of the human host response, indicated a significant therapeutic effect within two weeks of starting adjunctive phage therapy. These findings strongly suggested the release of bacterial cells or cell fragments into the bloodstream from deep bony infection sites early in treatment. This was associated with transient fever and local pain and with evidence of marked upregulation of innate immunity genes in the host transcriptome. Adaptive immune responses appeared to develop after a week of therapy and some immunomodulatory elements were also observed to be upregulated.

**Keywords** bacteriophage; child; immune response; osteoarticular infection; *Pseudomonas aeruginosa*

**Subject Categories** Autophagy & Cell Death; Immunology; Microbiology, Virology & Host Pathogen Interaction

## Introduction

Bacteriophages (phages), first used as antibacterial agents more than a century ago (Abedon *et al*, 2011), offer new hope for highly drug-resistant infections (Schooley *et al*, 2017; Petrovic Fabijan *et al*, 2020). A recent publication provided data on dosing, distribution kinetics and treatment response to adjunctive phage therapy in adult patients with overwhelming *Staphylococcus aureus* bacteraemia and septic shock (Petrovic Fabijan *et al*, 2020) but uncertainties remain about optimal dosing and treatment duration. Here we describe phage, bacterial and host responses to intravenous (IV) adjunctive phage therapy for a chronic osteoarticular *Pseudomonas aeruginosa* infection ($bla_{NDM-1}$-positive strain; Table 1) complicating the internal fixation of a traumatic fracture-dislocation in a 7-year-old girl. Prior surgical debridement and hardware removal and three months of IV antibiotics had failed to stop, the progress of infection (Fig 1A–D) and no other medical or surgical options were available.

## Results

### Clinical response to adjunctive phage therapy

Under informed consent, we administered 0.9 ml of $10^{11}$ plaque-forming units (PFU)/ml of Pa14NPΦPASA16 (PASA16) intravenously once (days 1, 2 and 4–7) or twice-daily (12 h apart; days 3 and 8–14) for two weeks, beginning three months into a twelve-month course of IV colistin and aztreonam. Strong lytic activity of PASA16 *in vitro* against a surgically obtained isolate of *Pseudomonas aeruginosa* (Ppa2.1; efficiency of plating [EOP] 1) was unaffected by these antibiotics *in vitro* (additive interaction).

Transiently increased heel pain 8 h after the first dose of PASA16 was followed by fever to 39.7°C several hours later (Fig 2).

1 Department of Infectious Diseases and Microbiology, The Children's Hospital at Westmead, Westmead, NSW, Australia
2 Sydney Medical School, The University of Sydney, Sydney, NSW, Australia
3 Centre for Infectious Diseases and Microbiology, Westmead Institute for Medical Research, Westmead, NSW, Australia
4 School of Medical Sciences, University of New South Wales, Sydney, NSW, Australia
5 Institute of Dental Sciences and School of Dental Medicine, The Hebrew University, Jerusalem, Israel
6 Department of Clinical Microbiology and Infectious Diseases, Hadassah-Hebrew University Medical Center, and the Faculty of Medicine, The Hebrew University, Jerusalem, Israel
7 Adaptive Phage Therapeutics, Gaithersburg, MD, USA
8 Department of Orthopaedics, The Children's Hospital at Westmead, Westmead, NSW, Australia
9 Westmead Hospital, Western Sydney Local Health District, Westmead, NSW, Australia
*Corresponding author. Tel: +61 298451902; E-mail: ameneh.khatami@health.nsw.gov.au
**Corresponding author. Tel: +61 286273411; E-mail: jonathan.iredell@sydney.edu.au

**Table 1.** Antimicrobial susceptibility profile of *Pseudomonas aeruginosa* isolated from a surgically obtained sample of bone from a paediatric patient with chronic infection.

| Antimicrobial agent | MIC (µg/ml) | EUCAST Interpretation |
|---|---|---|
| Colistin | 1 | S |
| Polymyxin B | 1 | – |
| Fosfomycin | ≥ 256 | R |
| Aztreonam | 4 | I |
| Aztreonam/Avibactam | 4/4 | – |
| Meropenem | 16 | R |
| Meropenem/Vaborbactam | 16/4 | R |
| Imipenem | > 32 | R |
| Imipenem-relebactam | > 32/4 | R |
| Eravacycline | 16 | R |
| Cefiderocol | 4 | S |

EUCAST, European Committee on Antimicrobial Susceptibility Testing; MIC, minimum inhibitory concentration; S, susceptible; R, resistant; I, intermediate.

Low-grade temperatures (37.7–37.8°C) were also recorded ∼6–12 h after the second and third doses of phage but none thereafter. C-reactive protein (CRP) peaked on day 4 (20.1 mg/l) (Fig 2) and a slight increase in serum Ig-G levels from baseline to day 30 (14.7 g/l to 16.1 g/l, normal range 6.24–14.4 g/l) was noted, with no change in serum complement levels (C3/C4). Pain-free weight bearing on the affected leg was achieved for the first time since the initial injury, seven weeks after completion of adjunctive phage therapy, and long-term follow-up demonstrated radiological improvement (Fig 1A–D). Five months after completion of all antimicrobial treatment the patient had no further infective symptoms, only occasional activity-related pain in the left ankle and was able to walk 7–8,000 steps per day. Moderate functional disability persists due to previous tissue damage, requiring ongoing surgical management and physiotherapy.

### Therapeutic monitoring of bacterial and phage kinetics in the bloodstream

Blood cultures yielded no bacteria at any stage, but viable phage was recovered from pre-dose samples up until day 5, consistent with productive infection of the target bacteria throughout the dosing interval (Fig 2). The initial plan to increase to twice-daily dosing after establishing tolerability over the first 2 days, was modified on the basis of this. The absence of viable phage on day 7, lead to increased phage dosing in the second week which appeared to be associated with further bacterial lysis, as suggested by detection of bacterial debris in the form of *Pseudomonas* DNA. Overall, the dynamics of bacterial and phage DNAemia (to ∼$10^5$ genome equivalents/ml) are most consistent with a classic "predator-prey" relationship (Fig 2).

### Immune response to adjunctive phage therapy in a child

Numerous differentially expressed (DE) genes were observed after background threshold normalisation and one-way ANOVA comparison of before (day 0; D0), to during (period 1 = D2, D4; period 2 = D5, D7; period 3 = D9, D11) and after adjunctive phage therapy (period 4 = D15, D29), with unadjusted $P < 0.05$. Initially (D2–4), there appeared to be strong upregulation of several genes linked to the innate immune response (GO:0045087), including *CD244, CD180, ATG5, CHUK, UBC, CLEC4A, TLR8, LY86, TLR7, TBK1, MARCO, IFI16, BST2, IRF7, DDX58* and *SERPING1* (Fig EV1). There were particularly strong signals linked to IFN pathways (e.g. *SERPING1* and *ISG15*, which modulate IFN-driven inflammatory responses) and to antiviral responses (e.g. *DDX5* encoding RIG-1, and *IRF7* which participates in a feedback loop with IFN-1 in the JAK-STAT pathway (Marie *et al*, 1998)).

Several of these DE transcripts persisted towards the end of the first week (D5–7), but some of the thirteen DE genes in this period (*TNFRSF17, CD1C, CD244*) are associated with the adaptive immune response (GO:002250), albeit not significantly after adjustment ($P = 0.004$ unadjusted). Many of the DE genes in the second week of therapy (period 3) had been evident since first starting adjunctive phage therapy and were mostly those linked to innate immunity (e.g., *CD244, IL5RA, CHUK, ATG5, CD180, TBK1*), some persisting as DE for two weeks after adjunctive phage therapy was completed (D15-29 vs. D0), when an additional 8 DE genes (*LRP1, IL3RA, ATF1, MAP2K4, HLA-DMB, CD1C, HLA-DPA1, HLA-DPB1*) were noted for the first time (Fig EV1).

Principal component analysis points to different gene profiles between periods of adjunctive phage therapy and significant differences in the expression of 18 genes associated with the innate immune response (Gene Ontology [GO]: 0045087, $P = 1.03 \times 10^{-11}$) that were most obviously differentially expressed between D0 (immediately prior to adjunctive phage therapy) and D2–4 (24–72 h after initiation of adjunctive phage therapy; $P < 0.05$). This coincided with the first infusions of phage, the subsequent bacterial DNAemia spike (Fig 2), fever, localised bone pain, and a rise in CRP and is generally consistent with the analysis above.

Hierarchical clustering analysis (HCL) further confirmed the enrichment of innate immune response-associated genes on D2, which appeared to subside over a few days (Fig 3A). While seven of these (*ATG5, CD180, CD244, PEZ1, IL5RA, PTGDR2* and *SMPD3*) appeared prominent even after completion of adjunctive phage therapy (D29), this was only significant up to D5–7 after application of *post hoc* statistical adjustment (Benjamini–Hochberg).

The most significant gene profile changes at systems level occurred within a few days, tapering off after the first week, so that by D29 the gene profile was similar to that of pre-therapy, as shown in the HCL analysis (Fig 3A). Gene Set Variation Analysis (GSVA; Hanzelmann *et al*, 2013) illustrates that the early upregulation of genes enriched for innate immune response in periods 1 (D2, D4) and 3 (D9, D11) are in synch with the kinetics of *Pseudomonas* DNAemia (Fig 2), followed from D5 onwards by upregulation of genes associated with the adaptive immune response (Fig 3B), as observed in the HCL analysis (Fig 3A) and as hinted at in the initial ANOVA comparison (above).

Visualisation of enriched and biologically significant pathways illustrates the significance of toll-like receptor signalling (Fig EV2A), autophagy (Fig EV2B) and JAK-STAT signalling pathways (Fig EV2C). Gene network visualisation (Wickham *et al*, 2019; Larsson, 2020; Pedersen, 2020) highlights a small group of genes as differentially expressed consistently which begins after the

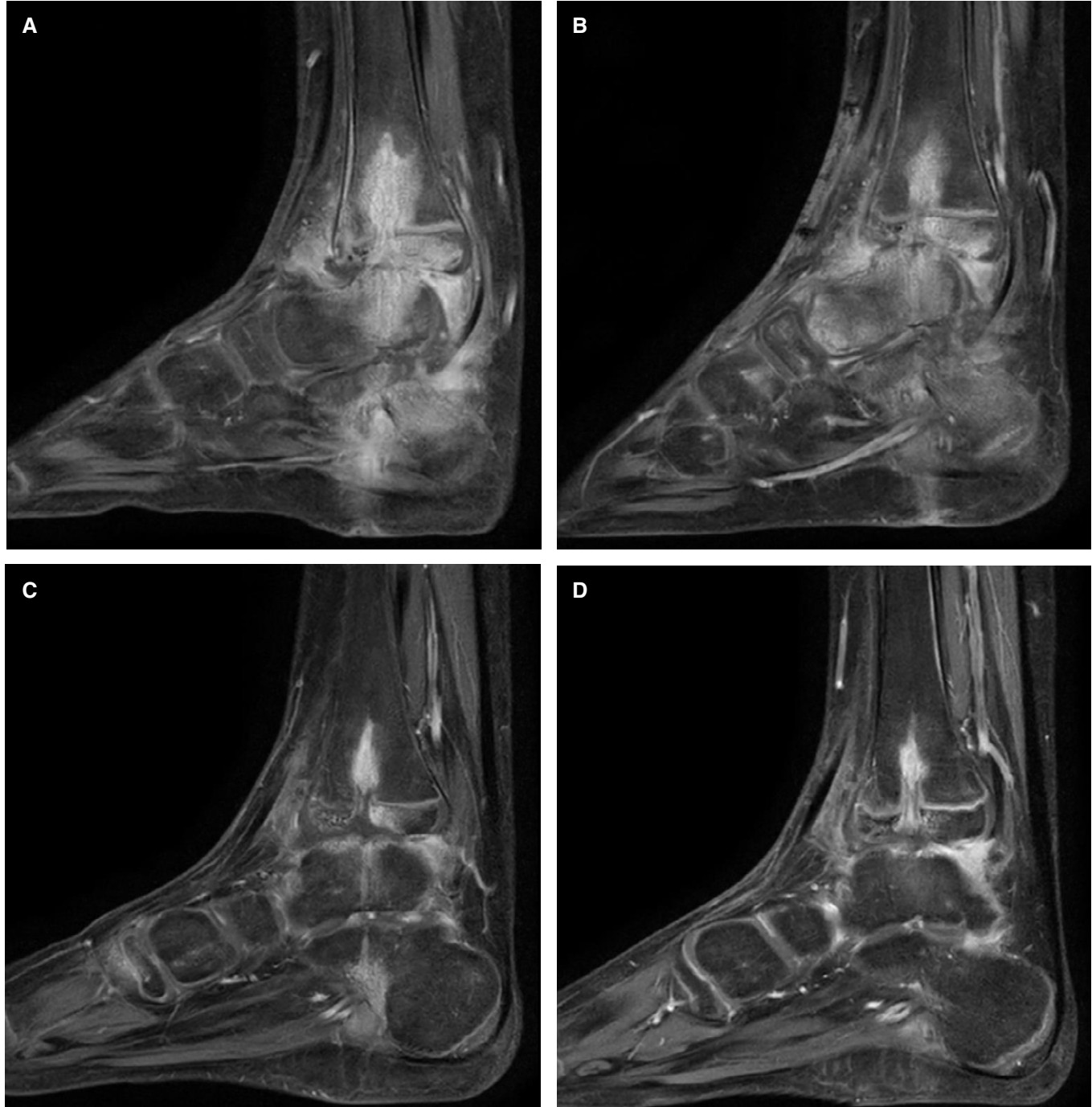

**Figure 1.** Gadolinium-enhanced magnetic resonance imaging of the lower leg of a paediatric patient with chronic *Pseudomonas aeruginosa* infection following surgical fixation (K-wire, removed), over 12 months of treatment and follow-up.

A   One week prior to beginning a 12 month course of intravenous (IV) antibiotics, showing extensive inflammation along fixation device tracts with bone marrow inflammation in the heel and the ankle joint itself and in adjacent muscles and tendons.

B   Two months after surgical debridement and initiation of IV colistin and aztreonam, four weeks prior to adjunctive phage therapy: apparent extension of inflammation in the foot and ankle.

C   After 5 months of IV colistin and aztreonam, 6 weeks after completion of a 2-week course of IV phage: marked reduction in the marrow oedema adjacent to surgical tracts and reduced bony inflammation in the foot and ankle overall.

D   After 12 months of IV colistin and aztreonam, eight months after completion of adjunctive phage therapy: continuing resolution is apparent.

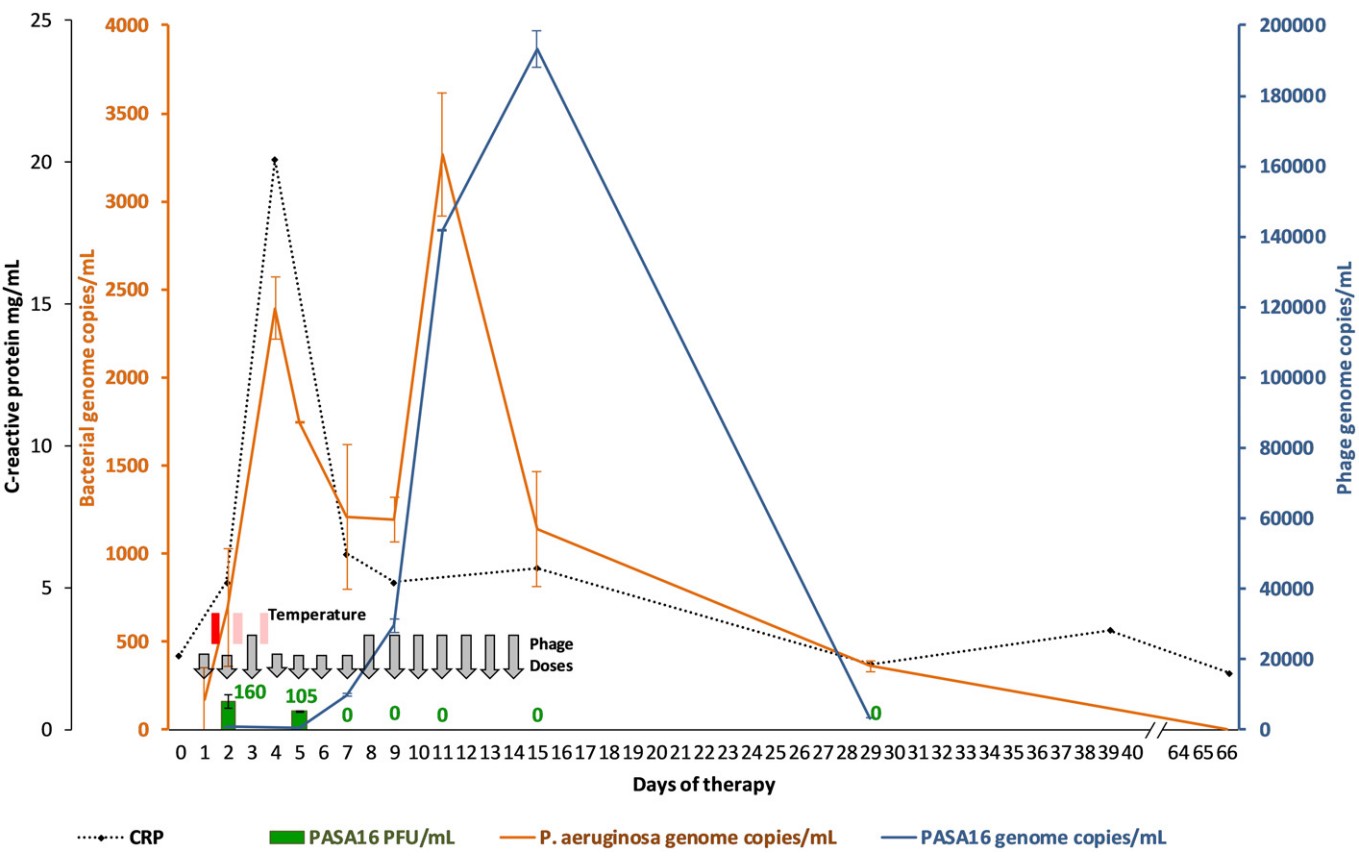

**Figure 2.** *Pseudomonas aeruginosa* and bacteriophage kinetics in blood over 2 weeks of adjunctive phage therapy and 7 weeks of follow-up.

PFU: plaque-forming units; CRP: C-reactive protein; phage doses given once (short) or twice (long vertical arrows) daily; coloured temperature boxes represent brief episodes of high-grade (red) or low-grade (pink) fever; blood samples obtained prior to morning phage dose; error bars represent high/low values from duplicate experiments; limit of detection for phage and bacterial quantitative PCR assay are 10 and 130 genome copies/ml, respectively; phage DNA detected on days 2 and 5 were ∼700 genome copies/ml.

Source data are available online for this figure.

commencement of adjunctive phage therapy and persists after its completion (Fig EV1). *CD180, CD244* and *ATG5* are strongly associated with the innate immune response (GO:0045087, *P* = 0.006) and also appear as key gene expression drivers in pathway enrichment analyses (Table 2). *CD180* and *ATG5* were upregulated while *IL5RA, SMPD3, PTGDR2* and *CD244*, also associated with the innate immune response, were consistently downregulated (Table 2). *FEZ1*, linked to intracellular viral trafficking (Haedicke *et al*, 2009), was consistently downregulated after adjunctive phage therapy was initiated, except for D5–7, when the spike in phage DNAemia coincided with its relative upregulation.

**Prophage content of *Pseudomonas aeruginosa* isolate (Ppa2.1)**

PHAge Search Tool Enhanced Release (PHASTER) algorithm identified 2 intact (complete) prophage sequences matching the F10-like (fam. *Siphoviridae*, 40.5 kb) and Pf1 (*Inoviridae*, 14.9 kb). In addition, PHASTER identified 5 prophage-like elements that scored < 90. Of these, sequence number 5 (Fig EV3) showed the highest score (87; highly probable prophage) matching the phiCTX-like phage (fam. *Myoviridae* 41.5Lb). These results were compared with

PHASTER analysis of reference strain PAO1 (accession number: NC_002516.2), suggesting similarity in prophage content between the two; both strains contain Pf-like *Inoviridae* phages.

# Discussion

We have previously described phage distribution kinetics and response to therapy in adult patients with *S. aureus* bacteraemia and septic shock (Petrovic Fabijan *et al*, 2020). Here we report, in contrast, phage attack directed against an archetypal Gram-negative pathogen, *P. aeruginosa*, deeply sequestered in bone. The surges of *Pseudomonas* DNA ($10^2$–$10^3$ genome copies/ml) into the bloodstream of the patient are expected to be accompanied by between 0.05 and 0.5 ng/ml (Jackson & Kropp, 1992; Eng *et al*, 1993) (∼40–400 EU/kg) of bacterial endotoxin. This is orders of magnitude higher than the endotoxin levels in the administered phage preparations, which were below the accepted human pyrogenic threshold of 5 EU/kg per dose, and thus would be expected to produce a strong inflammatory response, typically via Toll-like receptor 4 (TLR4) pathways. Increased phage dosing in the second week was

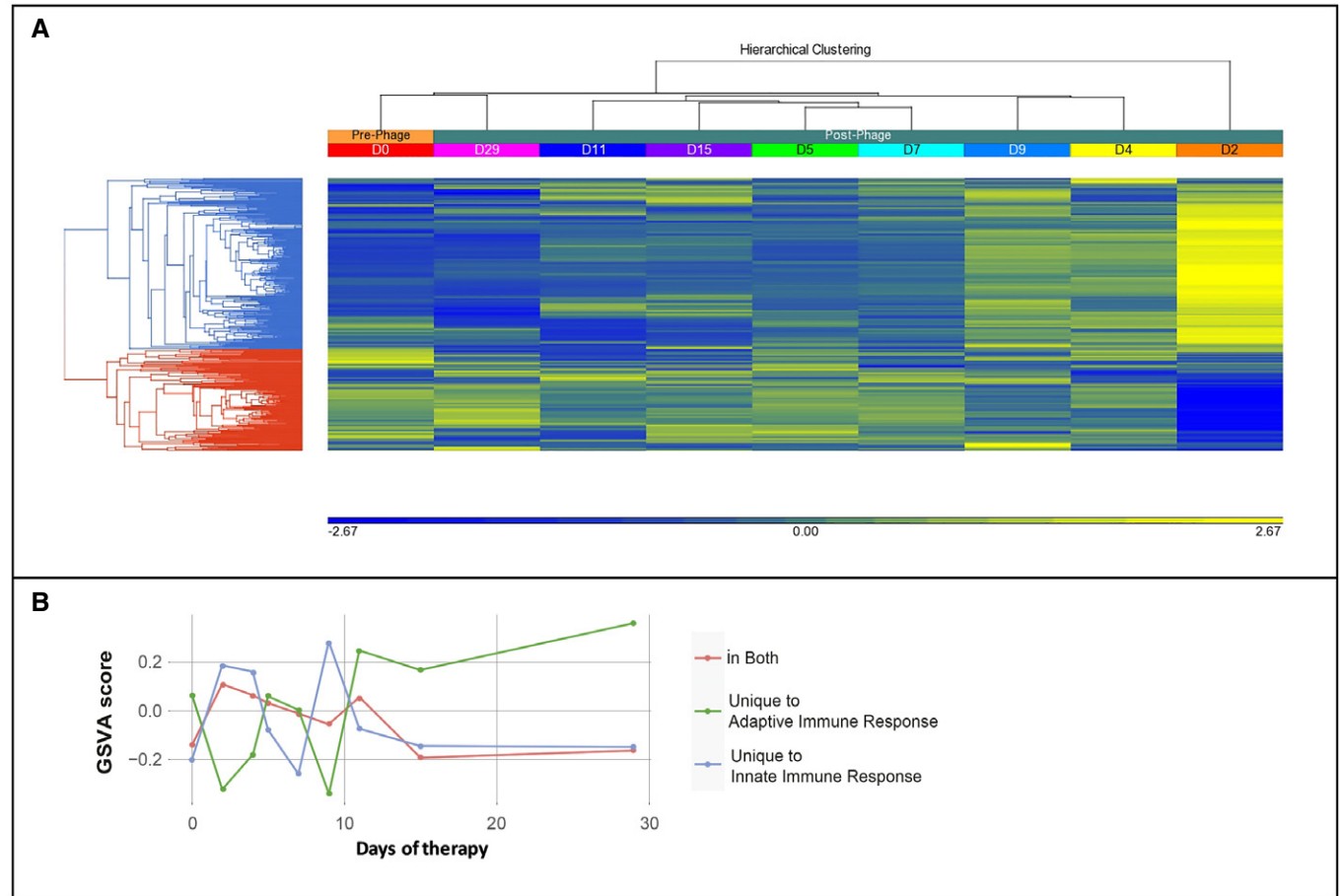

**Figure 3. Immune response gene expression profile (3A) and gene set variation analysis (3B) in a child receiving intravenous adjunctive phage therapy over 2 weeks with 2 weeks of follow-up.**

A  HCL analysis of gene expression for a child receiving intravenous phage therapy. D0: pre-phage, D2–D11: during phage administration, D15: one-day post-adjunctive phage therapy, D29: 15 days post-adjunctive phage therapy. Significantly upregulated genes ($n$ = 58, horizontal bar at bottom, yellow end) in D2–4 were enriched for innate immune response (adjusted $P$ = $1.03 \times 10^{-11}$), and genes enriched for adaptive immune responses were expressed from D5–7 onwards (unadjusted $P$ = 0.004, adjusted $P$ = 0.20), with D15–29 showing a similar profile to D0 (linkages on top). Hierarchical clustering of gene expression reflects concordantly up- or down-regulated genes expressed at different time-points, within the same regulatory cascade (blue, innate, red, adaptive immune responses) as determined by gene enrichment and pathway enrichment analyses (linkages on left).

B  Gene set variation analysis of genes enriched for innate immune response and adaptive immune response from D0 to D29 was based on REACTOME pathways accessed using MSigDBR. Overlapping genes from both lists were extracted as a separate gene set prior to GSVA analysis. This visualisation indicates a dichotomy of gene expression, with upregulation of the innate immune response initially (D2, D4) followed by an adaptive immune response developing towards the end of the first week.

associated with further bacterial lysis but the dynamics of bacterial and phage DNAemia (to $\sim 10^5$ genome equivalents/ml), similar to previous observations (Khawaldeh *et al*, 2011; Petrovic Fabijan *et al*, 2020) are most consistent overall with a classic "predator-prey" relationship (Fig 3), with predators (phage) thriving while prey (bacterial hosts) are plentiful but declining as prey populations are consumed. In contrast, it is unlikely that the detected *Pseudomonas* DNAemia is solely attributable to contamination from the phage infusion. The measured peaks of *Pseudomonas* DNA in the patient's serum were greater than what could be measured in administered vials of PASA16, and the biphasic rise in bacterial DNA did not coincide precisely with periods of constant dosing or dosing changes.

While we identified at least two genuine prophages (intact) and potentially five others (incomplete) within our patient's *P. aeruginosa* isolate (Ppa2.1), we did not observe maladaptive innate viral response nor impaired bacterial clearance, as previously reported to be triggered by Pf-like prophages (Sweere *et al*, 2019). In contrast, our data reveal the differential expression of genes linked to autophagy such as *ATG5*, that may be beneficial in clearance of *P. aeruginosa* infection (Yuan *et al*, 2012), and others (*CD180* and *CD244*), which may dampen the inflammatory response (Karper *et al*, 2013) during adjunctive phage therapy. CD244 is part of the signalling lymphocytic activation molecule (SLAM) family. It modulates activation and differentiation of a number of immune cells and interconnects both innate and adaptive immune responses and was

Table 2.  Candidate genes enriched for innate immune response pathway in a child receiving intravenous adjunctive phage therapy.

| Gene | Accession | P-value GO: 0045087 (Innate Immune Response) | Full name | Fold-change (Period 1 vs. D0) | P-value | Fold-change (Period 2 vs. D0) | P-value | Fold-change (Period 3 vs. D0) | P-value | Fold-change (Period 4 vs. D0) | P-value |
|---|---|---|---|---|---|---|---|---|---|---|---|
| CD244 | NM_016382.2 | 0.0338035 | CD244 molecule, NK cell receptor 2B4 | −1.8474 | 0.0050835 | −1.54048 | 0.0172098 | −1.54398 | 0.016914 | −1.41521 | 0.0344261 |
| ATG5 | NM_004849.2 | 0.0132674 | autophagy related 5 | 1.36845 | 0.0022089 | 1.24926 | 0.0077257 | 1.32142 | 0.0034276 | 1.30724 | 0.0039626 |
| CD180 | NM_005582.2 | 0.0008005 | CD180 molecule | 1.19037 | 0.0119963 | 1.41165 | 0.0009871 | 1.46428 | 0.0006696 | 1.72345 | 0.0001673 |
| IL5RA | NM_000564.3 | 0 | interleukin 5 receptor, alpha | −1.141 | 0 | −1.141 | 0 | −1.141 | 0 | −1.141 | 0 |
| SMPD3 | NM_018667.3 | 0 | sphingomyelin phosphodiesterase 3, neutral membrane (neutral sphingomyelinase II) | −1.9285 | 0 | −1.9285 | 0 | −1.9285 | 0 | −1.9285 | 0 |
| PTGDR2 | NM_004778.1 | 0.0011436 | prostaglandin D2 receptor 2 | −1.75981 | 0.0371006 | −2.58629 | 0.0066510 | −6.67246 | 0.0004961 | −7.0835 | 0.0004398 |
| FEZ1 | NM_005103.4 | 0.0000851 | fasciculation and elongation protein zeta 1 (zygin I) | −1.0625 | 0.0088444 | 1.18623 | 0.0001776 | −1.0625 | 0.008844 | −1.0625 | 0.008844 |

D0, immediately prior to adjunctive phage therapy; Period 1 = D2, D4, Period 2 = D5, D7, Period 3 = D9, D11 (during adjunctive phage therapy); Period 4 = D15, D29 (post-adjunctive phage therapy); NK cell, natural killer cell.

downregulated throughout adjunctive phage therapy. By contrast, *CD180*, in the family of pathogen receptors linked to natural killer (NK) cell-mediated cytotoxicity (KEGG: hsa04650), was consistently upregulated after starting adjunctive phage therapy (Table 1). TLR4 mediates B-cell recognition of lipopolysaccharide (LPS), a membrane constituent of Gram-negative bacteria that drives the inflammatory response in sepsis. While we could not find a significant direct correlation between bacterial load and specific gene signatures, upregulation of *CD180* likely is expected to result from the release of LPS into the system after phage-mediated bacterial lysis. *FEZ1*, linked to intracellular viral trafficking (Haedicke *et al*, 2009), appeared to rise with spikes in phage DNAemia (e.g. d5–7). By contrast, the significant upregulation of *ATG5*, involved in several cellular processes including downregulation of the innate antiviral immune response, hints at an immunomodulatory response to introduction of phage. ATG5 has a key role in autophagic vesicles and has been directly implicated in immunomodulatory effects in the course of other viral infections (Jounai *et al*, 2007), and its upregulation was consistent from immediately after therapy began (Table 1).

The kinetics of significant changes in Toll-like receptor signalling, autophagy and JAK-STAT signalling pathways align temporally with the clinical signs and symptoms and phage-bacterial kinetics and are consistent with the innate immune response that might be expected after phage-mediated bacterial lysis and the liberation of inflammatory bacterial mediators such as LPS. Similarly, an adaptive immune response to both bacteria and phage is to be expected after 7–10 days, and may be associated with absence of viable phage in serum beyond the first week of therapy as seen here and previously (Petrovic Fabijan *et al*, 2020). Although we cannot

exclude other factors influencing the immune responses noted, including the release of endogenous prophages (Secor *et al*, 2020), the close temporal coincidence with phage administration, suggests either a direct or indirect immunomodulatory effect that justifies closer scrutiny.

It appears that clinical signs and widely available measures such as C-reactive protein, together with quantitation of phage and bacterial dynamics, can be used to guide the therapeutic interaction of the three main actors: phage, bacteria, and human host. The kinetics of the response are similar to previous descriptions (Petrovic Fabijan *et al*, 2020) and indicate that most benefit accrues within the first ten days of treatment. Such multi-modal therapeutic phage monitoring is important to help us understand this intervention better.

## Materials and Methods

### Details of PASA16 phage used

Titer: $1.72 \times 10^{11}$ PFU/ml; volume/vial: 1.1 ml; endotoxin content: 170 EU/ml.

*Pseudomonas aeruginosa* DNA content: $\sim 10^6$ genome copies/ml. This equates to $\sim 10^3$ genome copies/ml after dilution in the patient's circulating blood volume (80 ml/Kg).

### Quantification of PASA16 phage in patient serum

Viable phage was recovered from serum onto a lawn of an aztreonam- and colistin-resistant strain of *P. aeruginosa* (JIP697;

EOP 1) as previously described (Clokie & Kropinski, 2009). Quantification of recovered phage was undertaken using 100 µl of undiluted and diluted serum (1:10 in SM buffer) co-incubated with 300 µl of log-phase bacteria embedded in soft-agar, also as previously described (Clokie & Kropinski, 2009). Visible plaques were counted and multiplied by the dilution factor to obtain PFU/ml. Experiments were performed in duplicate.

### Quantification of PASA16 and Pseudomonas genomes in patient blood and serum and Pseudomonas DNA in PASA16 vials administered to the patient

DNA extracted from serum was used for quantitative (q)PCR targeting a conserved hypothetical gene in PASA16 (F: 5'-GCG AGT CCA GGT CCA ACT AC-3'; R: 5'-GTT GCA TAT CGC CCA GCT TG-3'). Reaction volumes were 6.25 µl QuantiNova SYBR® Green PCR Kit (Qiagen), 1.25 µl of each primer (1 mM), 0.75 µl water and 3 µl of DNA template. PCR conditions were two-minute initial denaturation at 95°C, then 45 cycles of 95°C for 5 s, 60°C for 10 s and 72°C for 10 s. Reaction efficiency was 89.8% with $R^2$ of 0.9985. DNA from whole blood (in PAXgene® blood RNA tubes; BD Diagnostics, Franklin Lakes, New Jersey, USA) was extracted using a commercial kit (QIAamp DNA Blood Mini Kit®; Qiagen, Hilden, Germany) with previously described modifications (Petrovic Fabijan *et al*, 2020), and quantified using 16-well multiplexed-tandem real-time PCR targeting *Pseudomonas* 16S rDNA (custom-designed, AusDiagnostics, Mascot, Australia) according to manufacturer instructions (reaction efficiency 93.1%, $R^2$ 0.9983). Standard curves were generated by spiking 10-fold serial dilutions of log-phase *P. aeruginosa* and PASA16 into whole human blood (*P. aeruginosa*, $1.3–1.3 \times 10^6$ colony-forming units/ml) or serum (phage, $1 \times 10^{1–7}$ PFU/ml). All experiments were performed in duplicate.

### Whole genome sequencing of *Pseudomonas aeruginosa* isolate (Ppa2.1)

DNA from overnight bacterial culture was extracted using a DNeasy blood and tissue kit (Qiagen). Paired-end multiplex libraries were prepared using the Illumina Nextera XT kit. Whole genome sequencing was performed by the Antimicrobial Resistance Laboratory, Microbial Genomics Reference Laboratory (Centre for Infectious Diseases and Microbiology Laboratory Services, Institute of Clinical Pathology and Medical Research, Westmead Hospital, NSW, Australia) using the Illumina NextSeq 500 platform. Data were analysed using a modification of the Nullarbor bioinformatic pipeline (Seemann *et al*), incorporating analysis of contigs in PHASTER (PHAge Search Tool Enhanced Release; http://phaster.ca/) and using *P. aeruginosa* PAO1 (NC_002516.2) as a reference. Raw sequence reads are available on NCBI under the BioProject accession number PRJNA733000.

### Calculation of endotoxin levels associated with Pseudomonas DNA in patient blood

$10^2–10^3$ *Pseudomonas* genome equivalents/ml corresponds to 0.05–0.5 ng/ml of endotoxin [[Jackson & Kropp, 1992; Eng *et al*, 1993]] × 10 EU/ng × 80 ml/Kg blood volume = 40–400 EU/kg endotoxin.

### Determination of gene expression profiles (transcriptomics) during adjunctive phage therapy

RNA was extracted from whole blood (PAXgene® tubes) using a commercial kit (PAXgene Blood RNA Kit®, Qiagen) to study human gene expression using the Nanostring nCounter® system with the PanCancer immune panel (NanoString Technologies, Seattle, Washington, USA), as previously described (Petrovic Fabijan *et al*, 2020). Differentially expressed genes were identified after background threshold normalisation carried out to a panel of housekeeping genes and positive and negative probes ($n = 40$), with genes that were below the threshold filtered out. RCC files were imported using nSolver 4.0 (Versions 4.0.66, 4.0.70). Data were $\log_2$ transformed using Partek Genomics Suite (v7.19.1125), and one-way ANOVA was used to compare differential gene expression before (day 0), during (days 2–11; period 1 = D2, D4; period 2 = D5, D7; period 3 = D9, D11) and after adjunctive phage therapy (period 4 = D15, D29). To visualise gene expression changes from D0 to D29, hierarchical clustering analysis was carried out where expression normalisation was undertaken by shifting genes to mean of zero and scaled to standard deviation of one. Differential expression was represented as upregulation or downregulation on a coloured scale.

### Visualisation of gene and pathway enrichment analyses

Gene Set Enrichment Analysis and Pathway enrichment analysis (KEGG, https://www.genome.jp/kegg/) were calculated based on EASE Score with a modified Fisher exact *P*-value (The Database for Annotation, Visualization and Integrated Discovery (Jiao *et al*, 2012) v6.8, https://david.ncifcrf.gov/home.jsp). Visualisation of enriched and biologically significant pathways was carried out using PathView (Luo & Brouwer, 2013) where genes with fold change > 2 and < −2 were integrated into KEGG pathways. Gene enrichment analyses were based on Gene Ontology (GO) terms for innate and adaptive immune responses with *post hoc* Benjamini–Hochberg adjustment of *P*-values (Lin *et al*, 2010). Differentially expressed genes during adjunctive phage therapy were also visualised in a molecular gene network (Wickham *et al*, 2019; Larsson, 2020; Pedersen, 2020). Significant genes ($P < 0.05$) were mapped with gene expression fold changes, and differential expression was represented as upregulation or downregulation on a coloured scale. Gene set variation analysis (GSVA) was carried out to further dissect genes enriched for innate immune response and adaptive immune response from D0 to D29 (Hanzelmann *et al*, 2013). This was based on REACTOME pathways accessed using MSigDBR (Dolgalev, 2020). Overlapping genes from both lists were extracted as a separate gene set prior to GSVA analysis.

## Data availability

Bacteriophage Pa14NPΦPASA16 sequence data have been deposited in GenBank with the accession code MT933737.1 (http://www.ncbi.nlm.nih.gov/nuccore/MT933737.1/). Raw sequence reads from the clinical isolate of *Pseudomonas aeruginosa* (Ppa2.1) are available on NCBI under the BioProject accession number PRJNA733000 (https://www.ncbi.nlm.nih.gov/sra/PRJNA733000). All other data supporting the findings of this study are available within the paper

**The paper explained**

**Problem**

With increasing antimicrobial resistance, clinicians are faced with difficult-to-treat infections that pose a significant burden to individual patients and the health system. Phage therapy has been proposed as an alternative, non-antibiotic treatment for such infections but due to limited access, few data exist on its clinical use, especially in children.

**Results**

Multi-modal therapeutic monitoring of patients receiving phage therapy demonstrates the interaction between phage and bacteria is associated with complex human host responses to both bacteria and phage that directly influence the inflammatory syndrome.

**Impact**

Accumulating evidence from patients treated and closely monitored will allow increased understanding of this novel therapeutic option as a precision medicine tool to combat growing antimicrobial resistance.

or as supplementary Source Data files. Any other additional data that may be of interest are available from the corresponding author upon reasonable request.

**Expanded View** for this article is available online.

## Acknowledgements

We would like to acknowledge the patient and her family who kindly gave us permission to report these findings. Written informed consent was obtained from the child's mother, as well as approval from the Sydney children's Hospital Network Human Research Ethics Committee. All experiments conformed to the principles set out in the WMA Declaration of Helsinki and the Department of Health and Human Services Belmont Report. We would also like to thank Jessica Sacher and Jan Zheng from Phage Directory, Atlanta, GA, United States for helping our team source adjunctive phage therapy for this patient, as well as staff from the Westmead Institute for Medical Research core facility (Bioinformatics), supported by the Westmead Research Hub, the Cancer Institute New South Wales, the National Health and Medical Research Council and the Ian Potter Foundation, including Brian Gloss for his assistance in gene networks and pathway visualisations and Joey Lai for technical support in Nanostring experimets, Andrew Ginn from the Antimicrobial Resistance Laboratory, Institute of Clinical Pathology and Medical Research, Westmead Hospital, NSW, Australia for providing genomic sequencing data for the clinical isolate of *Pseudomonas aeruginosa* (Ppa2.1) and Ali Khalid from the Centre for Infectious Diseases and Microbiology, Westmead Institute for Medical Research, and Faculty of Medicine and Health at University of Sydney, Australia for his help in creating the graphical abstract for the manuscript. Bacteriophage Pa14NPΦPASA16 was isolated and characterised by the Hazan laboratory (Hebrew University, Jerusalem, Israel) and prepared for the treatment of this patient by Adaptive Phage Therapeutics (Maryland, USA) on compassionate grounds with no commercial value to The Children's Hospital at Westmead. This work was supported by grants to JI from the Australian National Health and Medical Research Council (APP1197534: Positive Solutions for Critical Infection).

## Author contributions

AK conceived the project, collated the data and prepared the manuscript. AK and JRI directed phage treatment of the patient and interpreted all clinical and laboratory data. RCYL designed the Nanostring pipeline and analysed and interpreted the transcriptomic data. AP-F processed patient samples, performed DNA/RNA extractions, conducted and interpreted phage susceptibility, phage-antibiotic *in vitro* interactions, initial Nanostring experiments, phage and bacterial kinetics and *in silico* prophage detection and characterisation. AK, JRI, RCYL and AP-F critically revised the manuscript. SA-O, RH and RN-P isolated and characterised the phage used to treat the patient. SA, PNB and QD oversaw overall clinical and orthopaedic management of the patient. MJB, JF and BAH provided guidance and advice during phage treatment. All authors reviewed the manuscript, contributed to revisions and approved the final version.

## Conflicts of interest

MB, JF and BH are employees of Adaptive Phage Therapeutics. All other authors declare that they have no conflicts of interest.

## For more information

- http://www.kidsresearch.org.au/about-kids-research
- https://phage.directory
- https://www.criticalinfection.com

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
