## [Review Process File · EMBO Molecular Medicine]

Bacterial lysis, autophagy and innate immune responses during adjunctive phage therapy in a child

Ameneh Khatami, Ruby Lin, Aleksandra Petrovic-Fabijan, Sivan Alkalay-Oren, Sulaiman Almuzam, Philip Britton, Michael Brownstein, Quang Dao, Joseph Fackler, Ronen Hazan, Bri'Anna Horne, Ran Nir-Paz, and Jonathan Iredell

DOI: [10.15252/emmm.202113936](https://doi.org/10.15252/emmm.202113936)

Corresponding authors: Ameneh Khatami (ameneh.khatami@health.nsw.gov.au) , Jonathan Iredell (jonathan.iredell@sydney.edu.au)

Review Timeline:

Submission Date:	12th Jan 21
Editorial Decision:	4th Mar 21
Revision Received:	30th May 21
Editorial Decision:	17th Jun 21
Revision Received:	15th Jul 21
Accepted:	19th Jul 21

Editor: Zeljko Durdevic

Transaction Report:

4th Mar 2021

Dear Dr. Iredell,

Thank you for the submission of your manuscript to EMBO Molecular Medicine, and please accept my apologies for the delay in getting back to you. We have received feedback from two of the three reviewers who agreed to evaluate your manuscript. Should referee #2 provide a report, we will send it to you, with the understanding that we will not ask for an additional revision. As you will see from the reports below, both referees find the study interesting and important. However, they also raise important criticism that I would like you to address in a major revision of the current manuscript. Furthermore, I would like you to consider publishing your manuscript as a scientific report (3 figures, ~22000 characters), for more information please check our "Author Guidelines".

<https://www.embopress.org/page/journal/17574684/authorguide#reportsarticleguide>.

Please also check our "Author Guidelines" for figure formatting, as some of the current figures could be presented as EV Figures.

<https://www.embopress.org/page/journal/17574684/authorguide#figureformat>

Addressing the reviewers' concerns in full will be necessary for further considering the manuscript in our journal. Please note that EMBO Molecular Medicine encourages a single round of revision only and therefore, acceptance or rejection of the manuscript will depend on the completeness of your responses included in the next, final version of the manuscript. For this reason, and to save you from any frustrations in the end, I would strongly advise against returning an incomplete revision.

We would welcome the submission of a revised version within three months for further consideration. However, we realize that the current situation is exceptional on the account of the COVID-19/SARS-CoV-2 pandemic. Please let us know if you require longer to complete the revision.

I look forward to receiving your revised manuscript.

Yours sincerely,

Zeljko Durdevic

***** Reviewer's comments *****

Referee #1 (Remarks for Author):

This report is concisely written, sometimes too much. The data collected sets the level of information needed to convince more clinicians on the safety and efficacy of adjunctive phage therapy. Have the following remarks/questions

The abstract is not reflecting what has been done, but what should be done in the future, unless I missed some details. Indeed the authors mentioned "real-time phage dose adjustments" based on phage and bacteria kinetics and host response. I doubt that all of the data were acquired within 24h to allow real-time adjustment. I believe that authors expect their data would be used to define a scheme of phage treatment, while so far these treatments were poorly designed with no justification of dose and frequency. Either adjust the abstract or given details in the results/methods section, how the results that you obtained along the treatment did affect its course.

Keep adjunctive phage therapy along the manuscript and perhaps propose to use APT to make it more easy to read. It is important to remind the readers that the antibiotic treatment was continuous during the treatment.

The last sentence of the abstract is highly hypothetical/speculative, which is lowering its impact. I understand that gene expression is only a first step towards understanding how the network of immune cells and signals is playing a role during this treatment, but too much hypothesis gives the feeling that data are not solid.

What is the volume in which the dose of 10^{11} PFU was administered. Was it done everyday at the same time? When 2 administrations were given, how much time between these?
What is the antibiotic resistance profile of the strain's patient ?

Figure 2 is difficult to understand and must redesigned. The important information is located time 18, the rest of the scale is not useful. Several MOI were tested (the last one is 100 and not 10) but all of them combined with antibiotics gave the same results ? Additive interaction is unclear. Basically, I cannot understand this figure and actually do you need a figure ?

Figure 3 is the most interesting and is pivotal but its design could be improved. The X axis between 42 and 62 could be cut in order to expand the size of the 0-42 section to enlarge it and improve the visualization. The only data that is not clear to me is why the DNA copies of phage genomes in blood is not detected during the first 4 days while PFUs were detected and then copies of phage DNA jumped from day 8 to day 16, while no PFU were detected. Also it seems that CRP was not measured at day 11, is any reason why ? At this time point both phage and bacterial DNA increased so CRP should also increase? Were 160 and 105 the exact number of PFUs in blood or is any power missing?

Figure 5 is also not so clear and I'm not convinced it is needed at all.

Figure 6a, at the bottom the legend for days is the same as the color code shown on the top with the hierarchy of the samples, please remove. On the other hand it is not clear what is the criteria splitting the left hierarchy in red and blue.

In the discussion the authors should mentioned that other indirect consequences of the treatment could explain some variations of immune response profiles. I'm referring to the work of the team of P. Bolyky that has shown that *P. aeruginosa* Pf prophage can manipulate the immune response. Within the same vein, bacterial debris released by *P. aeruginosa* could also induce the induction of prophage from the patient's strain. Did the authors sequenced this strain and checked whether it

has some prophages.

In the methods, what does mean plates with 30-300 viable plaques? How does it relate to the only PFU data shown on Figure 3 (160 and 105). The reference for phage titration is [ref3], which does not contain much information as it referred to ref [19-Petrovic] which finally explain that titration was done by serial dilutions plated on the lawn of bacteria and not by mixing phages and bacteria in soft-agar. Please resolve this discrepancy and cite an appropriate reference for the method you used.

In quantification of PASA16, the primer sequences are given. They are also repeated in the acknowledgements (remove it). Spiking should allow authors to determine the limit of detection that should be indicated in graphs or in legends.

In transcriptomics paragraph, change upregulation and down regulation by over- and down-expressed. Same remark in the next paragraph about visualization

Referee #3 (Comments on Novelty/Model System for Author):

This is a case report of a monitored treatment of an infection with phage. This approach is to be encouraged as it can yield a lot of useful data. I have put technical quality as medium because it is not well described so it is hard to assess. The novelty is high because the approach has not been used much before and looking for bacterial DNA as part of the monitoring I think is new. The medical impact could be high but because the work is unclearly described it is impossible to be sure that the conclusions are justified by the data.

Referee #3 (Remarks for Author):

It is great to see efforts being made to carry out carefully monitored single patient treatments with phage. The monitoring of bacterial DNA is as far as I know a new approach and could be very useful. I found the description of what was done hard to follow and there was important information missing.

In a description of a monitored treatment the things that I look for are:

1. Was the patient cured or improved. It is unclear whether they were cured, but the use of the leg was improved on at least one occasion and radiology improved. More clinical detail would be helpful.
2. Did the phage multiply? If it had then that is evidence of it working. There is no evidence that it multiplied and the very high doses used from the outset made it unlikely that multiplication would be detected. Phage counts in the blood fell from very low to undetectable quickly. Phage DNA was present but at a concentration well below that which could have been achieved from the redistribution of the administered doses.

I was initially excited by seeing the rise in bacterial DNA. This seemed to be evidence of bacterial lysis. However I do not know whether your phage preparation contained *P. aeruginosa* DNA as I can find no description of the method used for phage purification or testing done on it to look for bacterial DNA. The concentrations in the blood might be compatible with it all being DNA

administered with the phage. You might well be able to clear this point up now by testing a phage preparation prepared in the same way.

I am not an expert on the immunological aspects. It might be possible that some or possibly all of the reaction was to the phage preparation, and again more detail of how it was prepared and tested would be helpful.

To the Editors of EMBO Molecular Medicine

27th May 2021

Re: Bacterial lysis, autophagy and innate immune responses during phage therapy in a child

Dear Editors

Thank you for the opportunity to submit a revised version of our manuscript. We also thank the reviewers for their comments and suggestions and have responded to each of these below.

Additionally, we have amended the manuscript format to comply with the author instructions for a scientific report (20,672 characters, 3 figures, 2 EV figures, 2 tables).

Referee #1 comments:

This report is concisely written, sometimes too much. The data collected sets the level of information needed to convince more clinicians on the safety and efficacy of adjunctive phage therapy. Have the following remarks/questions

1. The abstract is not reflecting what has been done, but what should be done in the future, unless I missed some details. Indeed the authors mentioned "real-time phage dose adjustments" based on phage and bacteria kinetics and host response. I doubt that all of the data were acquired within 24h to allow real-time adjustment. I believe that authors expect their data would be used to define a scheme of phage treatment, while so far these treatments were poorly designed with no justification of dose and frequency. Either adjust the abstract or given details in the results/methods section, how the results that you obtained along the treatment did affect its course.

Response: Thank you for the comment. We were able to monitor for viable phage detectable in the patient's serum prior to each dose ("trough levels") with these results available within 24 hours. Prior to starting the treatment, we had planned to give once daily dosing for the first 2 days only to establish safety and tolerability, before switching to twice daily treatment from day 3. The presence of detectable viable phage 23 hours after dosing on day 2 suggested productive infection of the target bacteria was occurring throughout the dosing interval. We felt thus, that twice daily dosing was not required, and although the patient received twice daily dosing on day 3 (as shown in figure 3), which was in fact due to a nursing error where the evening dose was given early before our phage level results were available, we were able to adjust the dosing schedule to continue with once daily dosing. Further results on day 7 indicated that viable phage was no longer detectable in the patient's serum pre-dose and so we switched to twice daily dosing thereafter. The abstract and a section under the heading "Therapeutic monitoring of bacterial and phage kinetics in the bloodstream" have been modified slightly to make it more evident that phage and bacterial kinetics did in fact result in real-time dose adjustments for our patient.

2. Keep adjunctive phage therapy along the manuscript and perhaps propose to use APT to make it more easy to read. It is important to remind the readers that the antibiotic treatment was continuous during the treatment.

Response: we have used the term “adjunctive phage therapy” throughout as suggested, but have not used the abbreviation of APT, as this may be confused with Adaptive Phage Therapeutics who were the suppliers of the phage product for our patient.

3. The last sentence of the abstract is highly hypothetical/speculative, which is lowering its impact. I understand that gene expression is only a first step towards understanding how the network of immune cells and signals is playing a role during this treatment, but too much hypothesis gives the feeling that data are not solid.

Response: this sentence has been modified as suggested.

4. What is the volume in which the dose of 10^{11} PFU was administered. Was it done everyday at the same time? When 2 administrations were given, how much time between these?

Response: the volume administered per dose was 0.9 mL. This information has been added to the manuscript. As per usual inpatient care, intravenous medications are always given at approximately the same time each day with twice daily dosing occurring ~12 hours apart. We have added this information to the manuscript. There was however, the evening dose on day 3 which was actually given 5 hours early due to an error by nursing staff. We have not included this information due to restrictions on the manuscript length and because it is unlikely that this event had any significant impact to the course of treatment.

5. What is the antibiotic resistance profile of the strain's patient?

Response: The isolate of *P. aeruginosa* was positive for New Delhi metallo-beta-lactamase-1 (*bla_{NDM-1}*) and extensively drug resistant to all agents available for testing. The isolate was only susceptible to polymixin B (colistin) and ceftazidime (not registered in Australia at the time) and had intermediate susceptibility to aztreonam. This information has been added to the manuscript text and as a new table available in Extended View.

6. Figure 2 is difficult to understand and must be redesigned. The important information is located time 18, the rest of the scale is not useful. Several MOI were tested (the last one is 100 and not 10) but all of them combined with antibiotics gave the same results? Additive interaction is unclear. Basically, I cannot understand this figure and actually do you need a figure?

Response: Figure 2 has been removed as suggested.

7. Figure 3 is the most interesting and is pivotal but its design could be improved.
 - a. The X axis between 42 and 62 could be cut in order to expand the size of the 0-42 section to enlarge it and improve the visualization.

Response: This has been done as suggested

- b. The only data that is not clear to me is why the DNA copies of phage genomes in blood is not detected during the first 4 days while PFUs were detected and then copies of phage DNA jumped from day 8 to day 16, while no PFU were detected.

Response: In fact ~700 genome copies/mL of PASA16 (phage) were detected on days 2 and 5 (corresponding: 160 and 105 PFU/mL) but because the Y-axis for this goes up to 200,000 genome copies/mL (secondary Y-axis) it appears that no phage DNA was detected at the earlier time-points. A brief comment has been added to the figure legend to resolve this.

We also speculate that the development of adaptive immune responses in the second week of treatment may explain the absence of viable phage in the serum from day 7 onwards despite increasing phage DNAemia. A comment regarding this has been added to the discussion. Unfortunately we did not have the ability to measure circulating immune complexes that may have explained this phenomenon.

- c. Also it seems that CRP was not measured at day 11, is any reason why? At this time point both phage and bacterial DNA increased so CRP should also increase?

Response: Unfortunately the CRP measurement was missed on that day

- d. Were 160 and 105 the exact number of PFUs in blood or is any power missing?

Response: Each measurement for this graph was performed in duplicate and the midpoint value is reported. On day 2, 200 and 120 PFU/mL were observed by plaque assay (midpoint value 160 PFU/mL). On day 5, 110 and 100 PFU/mL were observed in the serum (midpoint value 105 PFU/mL).

8. Figure 5 is also not so clear and I'm not convinced it is needed at all.

Response: This figure has been deleted as suggested.

9. Figure 6a, at the bottom the legend for days is the same as the color code shown on the top with the hierarchy of the samples, please remove. On the other hand it is not clear what is the criteria splitting the left hierarchy in red and blue.

Response: Extra legend for days removed from the bottom of the figure as suggested. We have also added a brief comment in the figure legend to explain the left-hand hierarchical splitting.

10. In the discussion the authors should mentioned that other indirect consequences of the treatment could explain some variations of immune response profiles. I'm referring to the work of the team of P. Bollyky that has shown that *P. aeruginosa* Pf prophage can manipulate the immune response.

Response: Our patient had extensive chronic infection with *P. aeruginosa* for over 3 months prior to starting antimicrobial therapy, and had received over 3 months of targeted anti-pseudomonal antibiotics prior to administration of phage; however the immune signals noted were highly temporally associated with phage administration. Although we cannot exclude other factors influencing the immune response, it seems likely that phage administration, either

directly or indirectly, triggered these changes. The discussion has however been amended with reference to work from Bollyky's group.

11. Within the same vein, bacterial debris released by *P. aeruginosa* could also induce the induction of prophage from the patient's strain. Did the authors sequenced this strain and checked whether it has some prophages.

Response: The patient's isolate was sequenced and raw sequence reads from the clinical isolate of *Pseudomonas aeruginosa* are available on NCBI under the BioProject accession number PRJNA733000. Polished contigs were used to identify potential prophages. PHASTER (PHAge Search Tool Enhanced Release) program was able to identify 2 intact (complete) prophage sequences matching the F10-like (fam. *Siphoviridae*, 40.5Kb) and Pf1 (*Inoviridae*, 14.9Kb). In addition, PHASTER identified 5 prophage-like sequences that scored <90. Of these, sequence number 5 (Fig EV 3, new) showed the highest score (87), and probably corresponds to a third prophage identified in this isolate. These results were compared with PHASTER analysis of reference strain PAO1 (accession number: NC_002516.2), suggesting similarity in prophage contents between the two; both strains contain Pf1 *Inoviridae* phage that Sweere et al (Science, 2019) showed to trigger maladaptive innate viral pattern-recognition responses and impair bacterial clearance. We cannot exclude the contribution of prophage induction to the immune response but our data clearly show that both bacterial DNA spike and marked fever coincide with phage administration and the strong innate immune response observed on day 2 of phage therapy. Moreover, bacteria clearance seems not be impaired by either phage or prophages identified within patient isolate. Finally, the immune response is typical of a TLR-4 mediated response to bacterial endotoxin. Additional text has been included in the manuscript results and discussions regarding these findings.

12. In the methods, what does means plates with 30-300 viable plaques? How does it relates to the only PFU data shown on Figure 3 (160 and 105). The reference for phage titration is [ref3], which does not contain much information as it referred to ref [19-Petrovic] which finally explain that titration was done by serial dilutions plated on the lawn of bacteria and not by mixing phages and bacteria in soft-agar. Please resolve this discrepancy and cite an appropriate reference for the method you used.

Response: A different reference has been added here describing the double-layer plaque assay in detail and the text has been modified for greater clarity of methods.

13. In quantification of PASA16, the primer sequences are given. They are also repeated in the acknowledgements (remove it).

Response: Deleted as suggested.

14. Spiking should allow authors to determine the limit of detection that should be indicated in graphs or in legends.

Response: Limit of detection for assays added to the legend for figure 2 (previously figure 3) as requested.

15. In transcriptomics paragraph, change upregulation and down regulation by over- and down-expressed. Same remark in the next paragraph about visualization

Response: Thank you for this comment, but we have retained the use of up and down regulation to imply the relative differential expression, rather than over or under regulation which somehow also (especially for clinician readers) may imply an abnormal change.

Referee #3 (Comments on Novelty/Model System for Author):

1. This is a case report of a monitored treatment of an infection with phage. This approach is to be encouraged as it can yield a lot of useful data. I have put technical quality as medium because it is not well described so it is hard to assess. The novelty is high because the approach has not been used much before and looking for bacterial DNA as part of the monitoring I think is new. The medical impact could be high but because the work is unclearly described it is impossible to be sure that the conclusions are justified by the data.

Response: We thank the reviewer for their encouraging comments. We hope that the revisions made to the manuscript have improved clarity.

2. It is great to see efforts being made to carry out carefully monitored single patient treatments with phage. The monitoring of bacterial DNA is as far as I know a new approach and could be very useful. I found the description of what was done hard to follow and there was important information missing. In a description of a monitored treatment the things that I look for are:

- a. Was the patient cured or improved. It is unclear whether they were cured, but the use of the leg was improved on at least one occasion and radiology improved. More clinical detail would be helpful.

Response: We thank the reviewer for their encouraging comments. We have added a few more clinical details as requested.

- b. Did the phage multiply? If it had then that is evidence of it working. There is no evidence that it multiplied and the very high doses used from the outset made it unlikely that multiplication would be detected. Phage counts in the blood fell from very low to undetectable quickly. Phage DNA was present but at a concentration well below that which could have been achieved from the redistribution of the administered dose.

Response: As previously noted in many publications, including previous work from our team, phage administered intravenously is rapidly cleared from the circulation, usually in less than 2 hours. The persistence of viable phage in the serum of this patient 23 hours after an intravenous dose suggests productive infection of the target host with ongoing phage multiplication throughout the dosing interval (24 hours). We have mentioned this already in the manuscript.

Furthermore, during the second week of therapy when no viable phage was detected in the patient's serum, up to 10^5 genome copies/mL of phage DNA was detected in the serum 12 hours after the previous dose, whereas a dose of 10^{11} PFU/mL was administered as 0.9mL up to twice daily, which when distributed within the circulating blood volume of a paediatric patient (>2L) would have resulted in 10^8 PFU/mL. Given prior observations of rapid clearance of intravenously administered phage, this is also supportive of ongoing productive infection and phage multiplication within the *P. aeruginosa* host.

3. I was initially excited by seeing the rise in bacterial DNA. This seemed to be evidence of bacterial lysis. However I do not know whether your phage preparation contained *P. aeruginosa* DNA as I can find no description of the method used for phage purification or testing done on it to look for bacterial DNA. The concentrations in the blood might be compatible with it all being DNA administered with the phage. You might well be able to clear this point up now by testing a phage preparation prepared in the same way.

Response: The methods for production and purification of bacteriophages by the phage supplier (APT) are proprietary and not available for publication. However, we applied the same DNA extraction and PCR method we used for the patient's blood samples on the phage preparation and found $\sim 10^6$ genome copies/mL of *Pseudomonas* DNA, which would result in $\sim 10^3$ genome copies/mL after dilution in the patient's circulating blood volume, which is a little less than what we measured at the peak of *Pseudomonas* DNAemia. However to attribute the DNAemia to the infusion does not account for:

1. The biphasic rise in bacterial DNA - if the increase in DNA was from the phage product then the level of bacterial DNA in the patient's blood would have been constant over the course of treatment
2. The first peak of *Pseudomonas* DNAemia on day 4 coincides with an inflammatory response (rise in C-reactive protein)
3. The peaks in *Pseudomonas* DNAemia do not coincide exactly with dosing, e.g. increased to twice daily dosing on day 8 but bacterial DNA levels are virtually the same on day 7 and 9, and only peak again on day 11
4. the clear predator-prey curve (phage spike following bacterial spike)

The alternative explanation that fits the data well without requiring any coincidence or other unexplained biological phenomena is a classic predator-prey response, as is previously well described.

4. I am not an expert on the immunological aspects. It might be possible that some or possibly all of the reaction was to the phage preparation, and again more detail of how it was prepared and tested would be helpful.

Response: Thank you for your comment. Understandably, results from a single patient do not allow firm conclusions to be drawn and we have tried to stress the often speculative interpretation of our findings in the manuscript. As kindly noted by both reviewers, we believe this degree of therapeutic monitoring should be performed on all patients treated with phage to increase our understanding of this therapy over time. Unfortunately, we are not in a position to

provide further details regarding the preparation and testing of the phage product due to the proprietary nature of the data. Even though this is a limitation with respect to the research aspect of the work, as clinicians, we were satisfied regarding the sterility results and endotoxin levels of the product prior to administering for our patient.

Once again we thank the reviewers for their time and their constructive comments which have helped to improve our manuscript. We hope we have resolved all outstanding queries and look forward to hearing from you.

Kind regards,

Ameneh Khatami, on behalf of the authors

17th Jun 2021

Dear Dr. Khatami,

Thank you for the submission of your revised manuscript to EMBO Molecular Medicine. I am pleased to inform you that we will be able to accept your manuscript pending the following final amendments:

- 1) Please address all the referees' concerns. No additional experiments are required.
- 2) Figures and Tables: Main Figure and EV Figure legends should be at the end of the main manuscript file. Tables should be added to the final version of the manuscript file at the end of the file.
- 3) In the main manuscript file, please do the following:
 - Add callouts for Figure 1A-D and Figure EV1, EV2 and EV3.
 - Make sure that all special characters display well.
 - In M&M, include a statement that in addition that informed consent was obtained the experiments conformed to the principles set out in the WMA Declaration of Helsinki and the Department of Health and Human Services Belmont Report.
 - Rename "Competing Interest" to "Conflict of Interest".
 - Rename "Data and materials availability" to "Data availability".
- 4) Funding: Please make sure that information about all sources of funding are complete in both our submission system and in the manuscript. In addition to the grant institution add the grant number.
- 5) The Paper Explained: Please add it to the main manuscript text.
- 6) Synopsis:
 - Synopsis text: Please submit synopsis text as a separate .doc file. Check your synopsis text, revise it if necessary and submit final version with your revised manuscript. Please be aware that in the proof stage minor corrections only are allowed (e.g., typos).
 - Synopsis image: Please provide a striking image or visual abstract as a high-resolution jpeg file 550 px-wide x (250-400)-px high to illustrate your article.
- 7) For more information: There is space at the end of each article to list relevant web links for further consultation by our readers. Could you identify some relevant ones and provide such information as well? Some examples are patient associations, relevant databases, OMIM/proteins/genes links, author's websites, etc...
- 8) As part of the EMBO Publications transparent editorial process initiative (see our Editorial at <http://embomolmed.embopress.org/content/2/9/329>), EMBO Molecular Medicine will publish online a Review Process File (RPF) to accompany accepted manuscripts. This file will be published in conjunction with your paper and will include the anonymous referee reports, your point-by-point response and all pertinent correspondence relating to the manuscript. Let us know whether you agree with the publication of the RPF and as here, if you want to remove or not any figures from it prior to publication. Please note that the Authors checklist will be published at the end of the RPF.
- 9) Please provide a point-by-point letter INCLUDING my comments as well as the reviewer's reports and your detailed responses (as Word file).

I look forward to reading a new revised version of your manuscript as soon as possible.

Yours sincerely,

Zeljko Durdevic

***** Reviewer's comments *****

Referee #1 (Remarks for Author):

Authors have well revised their manuscript that now contains additional information. I have indicated below some points to be considered.

In the abstract the second sentence is far too long, please revise.

Last sentence of the introduction change "disease" to "infection"

In the Results, end of first paragraph, change 100% for 1, as EOP values are obtained from a ratio between two numbers (same change to be made in the method).

Main point: in the "therapeutic monitoring..." section, please state more clearly that PA genomes copies/mL means bacterial debris because there is no viable bacteria. The word debris will make more clear that DNA copies are related to bacterial lysis by the phage.

Title of fig 2: add the word "adjunctive" therapy

Legend of fig 2: remove "shaded" and replace "dark" by red and "light" by pink.

In the discussion

Even if the precise description of the phage preparation cannot be disclosed, the level of endotoxin of this product should be placed in the methods and not only in the discussion.

The two sentences about endotoxin levels could be more clear. I would suggest to start with DNA levels in the blood > endotoxin levels expected > compared to endotoxin levels of the phage administered.

Add the units for the amount of Pseudomonas DNA

Last paragraph, first sentence I suggest to remove "simple" and replace "kinetics" by "dynamics".

Referee #3 (Comments on Novelty/Model System for Author):

I have put medium for technical quality as one of the claims of the paper is that bacterial DNA detected was as a result of lysis of the bacteria in the patient by phage. It is also possible that at least some of the DNA arose from the phage preparation. I suggested in my previous review that the phage preparation could be tested for bacterial DNA content. This does not appear to have been done. Nor has this issue been addressed in the text.

Referee #3 (Remarks for Author):

Many of the points I raised previously have been adequately addressed. The main problem with this

paper is that in the absence of evidence of phage multiplication, a key piece of evidence that the phage may have lysed bacteria is that bacterial DNA rose when phage was given. Since phage is made by lysing bacteria it would be reassuring to know that the phage preparation did not contain sufficient bacterial DNA to account for some or all of the rise. Can that be tested? Are you confident that the purification methods used would have eliminated sufficient of the bacterial DNA to make it unlikely that the changes you saw were due to administered DNA? If so what methods did they use? I imagine it might even be possible for some bacterial DNA to become trapped within the phage particles.

My feeling is that the paper is hard to read and could be much shorter with fewer and less complicated looking figures. I understood little of the host response data and am unclear what the word autophagy means in this context. I think you have deleted the old fig 2-I am unclear. In my view it should be deleted.

To the Editors of EMBO Molecular Medicine

12th July 2021

Re: Bacterial lysis, autophagy and innate immune responses during phage therapy in a child

Dear Zeljko Durdevic

Thank you for the opportunity to submit a revised version of our manuscript. We also thank the reviewers for their comments and suggestions and have responded to each of these below. Additionally, we have amended the manuscript format as requested.

Editor Comments:

1) Please address all the referees' concerns. No additional experiments are required.

Response: Thank you, this has been done.

2) Figures and Tables: Main Figure and EV Figure legends should be at the end of the main manuscript file. Tables should be added to the final version of the manuscript file at the end of the file.

Response: This has been done

3) In the main manuscript file, please do the following:

- Add callouts for Figure 1A-D and Figure EV1, EV2 and EV3.

- Make sure that all special characters display well.

- In M&M, include a statement that in addition that informed consent was obtained the experiments conformed to the principles set out in the WMA Declaration of Helsinki and the Department of Health and Human Services Belmont Report.

- Rename "Competing Interest" to "Conflict of Interest".

- Rename "Data and materials availability" to "Data availability".

Response: All of these changes have been made as requested.

4) Funding: Please make sure that information about all sources of funding are complete in both our submission system and in the manuscript. In addition to the grant institution add the grant number.

Response: This has been done

5) The Paper Explained: Please add it to the main manuscript text.

Response: This has been done

The authors performed the requested editorial changes.

Reviewer Comments

Referee #1 (Remarks for Author):

Authors have well revised their manuscript that now contains additional information. I have indicated below some points to be considered.

1) In the abstract the second sentence is far too long, please revise.

Response: This has been revised as suggested.

2) Last sentence of the introduction change "disease" to "infection"

Response: Changed as suggested.

3) In the Results, end of first paragraph, change 100% for 1, as EOP values are obtained from a ratio between two numbers (same change to be made in the method).

Response: Changed as suggested.

4) Main point: in the "therapeutic monitoring..." section, please state more clearly that PA genomes copies/mL means bacterial debris because there is no viable bacteria. The word debris will make more clear that DNA copies are related to bacterial lysis by the phage.

Response: This has been modified as suggested with the addition of the highlighted text - The absence of viable phage on day 7, lead to increased phage dosing in the second week which appeared to be associated with further bacterial lysis, **as suggested by detection of bacterial debris in the form of *Pseudomonas* DNA.**

5) Title of fig 2: add the word "adjunctive" therapy

Response: Changed as suggested.

6) Legend of fig 2: remove "shaded" and replace "dark" by red and "light" by pink.

Response: Changed as suggested.

7) Even if the precise description of the phage preparation cannot be disclosed, the level of endotoxin of this product should be placed in the methods and not only in the discussion.

Response: Details of the phage product including vial volume, titer and endotoxin levels have now been added to the *Materials and Methods* section.

8) The two sentences about endotoxin levels could be more clear. I would suggest to start with DNA levels in the blood > endotoxin levels expected > compared to endotoxin levels of the phage administered.

Response: These sentences have been rewritten as suggested.

9) Add the units for the amount of *Pseudomonas* DNA

Response: This has now been added (genome copies/mL)

10) Last paragraph, first sentence I suggest to remove "simple" and replace "kinetics" by "dynamics".

Response: Changes made as suggested.

Referee #3 (Comments on Novelty/Model System for Author):

I have put medium for technical quality as one of the claims of the paper is that bacterial DNA detected was as a result of lysis of the bacteria in the patient by phage. It is also possible that at least some of the DNA arose from the phage preparation. I suggested in my previous review that the phage preparation could be tested for bacterial DNA content. This does not appear to have been done. Nor has this issue been addressed in the text.

Response: Thank you for this question again. As we had noted in our previous response, we applied the same DNA extraction and PCR method we used for the patient's blood samples on the phage

preparation and found $\sim 10^6$ genome copies/mL of *Pseudomonas* DNA, which would result in $\sim 10^3$ genome copies/mL after dilution in the patient's circulating blood volume, which is a little less than what we measured at the peak of *Pseudomonas* DNAemia. However to attribute the DNAemia to the infusion does not account for:

1. The biphasic rise in bacterial DNA - if the increase in DNA was from the phage product then the level of bacterial DNA in the patient's blood would have been constant over the course of treatment
2. The first peak of *Pseudomonas* DNAemia on day 4 coincides with an inflammatory response (rise in C-reactive protein)
3. The peaks in *Pseudomonas* DNAemia do not coincide exactly with dosing, e.g. increased to twice daily dosing on day 8 but bacterial DNA levels are virtually the same on day 7 and 9, and only peak again on day 11
4. the clear predator-prey curve (phage spike following bacterial spike)

The alternative explanation that fits the data well without requiring any coincidence or other unexplained biological phenomena is a classic predator-prey response, as is previously well described.

A brief version of this explanation has now been added to the text of the manuscript (see *Discussion* and *Materials and Methods* sections).

Referee #3 (Remarks for Author):

1) Many of the points I raised previously have been adequately addressed. The main problem with this paper is that in the absence of evidence of phage multiplication, a key piece of evidence that the phage may have lysed bacteria is that bacterial DNA rose when phage was given. Since phage is made by lysing bacteria it would be reassuring to know that the phage preparation did not contain sufficient bacterial DNA to account for some or all of the rise. Can that be tested? Are you confident that the purification methods used would have eliminated sufficient of the bacterial DNA to make it unlikely that the changes you saw were due to administered DNA? If so what methods did they use? I imagine it might even be possible for some bacterial DNA to become trapped within the phage particles.

Response: Thank you. As previously explained in our responses the methods for production and purification of bacteriophages by the phage supplier (APT) are proprietary and not available for publication. Even though this is a limitation with respect to the research aspect of the work, as clinicians, we were satisfied regarding the sterility results and endotoxin levels of the product prior to administering for our patient. However, we have responded to this question to the best of our ability above, and with modifications to the text of the manuscript which we hope will satisfy the reviewers.

2) My feeling is that the paper is hard to read and could be much shorter with fewer and less complicated looking figures. I understood little of the host response data and am unclear what the word autophagy means in this context. I think you have deleted the old fig 2-I am unclear. In my view it should be deleted.

Response: Thank you for the comments. We hope the additional explanations and revisions to the text (also suggested by reviewer 1) have made the manuscript easier to read. We have deleted the original fig 2 and fig 5 as previously suggested.

Once again we thank the reviewers for their time and their constructive comments which have helped to improve our manuscript. We hope we have resolved all outstanding queries and look forward to hearing from you.

Kind regards,

Ameneh Khatami, on behalf of the authors

We are pleased to inform you that your manuscript is accepted for publication and is now being sent to our publisher to be included in the next available issue of EMBO Molecular Medicine.

Corresponding Author Name: Ameneh Khatami and Jonathan Iredell

Manuscript Number: EMM-2021-13936